# Differential Expression of Insulin Growth Factor 1 (IGF-1) Isoforms in Different Types of Endometriosis: Preliminary Results of a Single-Center Study

**DOI:** 10.3390/biom14010007

**Published:** 2023-12-20

**Authors:** Nikolaos Blontzos, Despoina Mavrogianni, Konstantinos Ntzeros, Nikolaos Kathopoulis, Athanasios Moustogiannis, Anastassios Philippou, Michael Koutsilieris, Athanasios Protopapas

**Affiliations:** 1Endoscopic Surgery Unit, 1st Department of Obstetrics Gynecology, Medical School, National and Kapodistrian University of Athens, 11528 Athens, Greece; nikolas.mplontzos@yahoo.gr (N.B.); prototha@otenet.gr (A.P.); 2Experimental Laboratory, 1st Department of Obstetrics Gynecology, Medical School, National and Kapodistrian University of Athens, 11528 Athens, Greece; depy.mavrogianni@yahoo.com (D.M.); kntzeros@gmail.com (K.N.); 3Department of Physiology, Medical School, National and Kapodistrian University of Athens, 11528 Athens, Greece; amoustog@med.uoa.gr (A.M.); tfilipou@med.uoa.gr (A.P.); mkoutsil@med.uoa.gr (M.K.)

**Keywords:** endometriosis, endometriotic nodules, insulin growth factor 1 isoforms

## Abstract

Endometriosis is a benign, estrogen-dependent gynecological condition with an uncertain exact pathogenetic mechanism. The aim of this study was to evaluate the potential differential expression of Insulin Growth Factor 1 (IGF-1) isoforms in deeply infiltrating endometriotic (DIE) lesions, in ovarian endometriomas, and in the eutopic endometrium of the same endometriosis patients and to compare their expression with that in the eutopic endometrium of women without endometriosis. A total of 39 patients were included: 28 with endometriosis, of whom 15 had endometriomas only, 7 had DIE nodules only, and 6 had both DIE and endometriomas, and 11 without endometriosis served as controls. We noticed a similar pattern of expression between IGF-1Ea and IGF-1Ec, which differed from that of the IGF-1Eb isoform, possibly implying differential biological actions of different isoforms in DIE subtypes. We observed a tendency of lower expression of IGF-1Ea and IGF-1Ec in endometriomas without DIE compared to endometriomas with concurrent DIE or in DIE nodules. In conclusion, differential expression of IGF-1 isoforms may indicate that DIE with its associated ovarian lesions and simple ovarian endometriosis should be considered as two forms of the disease developing under different molecular pathways.

## 1. Introduction

Endometriosis is a benign, estrogen-dependent gynecological condition defined as the abnormal presence of endometrial stromal and glandular cells outside the endometrial cavity [1]. It is estimated that endometriosis affects up to 10% of women of reproductive age and, in much higher proportions, infertile women and women suffering from chronic pelvic pain [1]. Endometriosis represents a heterogeneous clinical entity that could be categorized into three distinct subtypes: ovarian endometriomas, superficial peritoneal implants, and endometriotic nodules deeply infiltrating the retroperitoneal space [2,3].

Despite extensive research and numerous theories, the exact pathogenetic mechanism of the disease remains uncertain. Several immunological, growth, and angiogenetic factors produced locally in the peritoneal cavity have been investigated as being involved in the pathophysiological process [4,5]. Insulin-like growth factor-1 (IGF-1) has been proven to play an integral role in the autocrine-paracrine regulation of ectopic endometrial cell growth and maintenance on the grounds of its anti-apoptotic and mitogenic actions [6,7,8].

In humans, the IGF-1 gene is in the long arm of chromosome 12 (12q22-24,1) and contains 6 exons separated by 5 introns. Alternative splicing during gene expression allows coding of three isoforms: IGF-1Ea, IGF-1Eb, and IGF-1Ec [9]. The first isoform, IGF-1Ea, is similar to the hepatic endocrine type of IGF-1. It is the most abundant isoform in the liver and contributes to growth hormone-dependent and growth hormone-independent secretion of IGF-1 into the circulation [9,10]. The IGF-1Eb isoform has been less studied, but it is thought to be predominantly expressed in the liver [9,10]. The third isoform, IGF-1Ec, has been initially investigated in the liver and skeletal muscle and is also named mechano growth factor (MGF) because its expression is increased after muscle tissue has been subjected to mechanical loading and/or damage (mechano-sensitive response) [11]. The exact physiological actions of these isoforms in the different subtypes of endometriosis have been poorly investigated [12].

Therefore, the aim of this study was to evaluate the potential differential expression of IGF-1 isoforms in deeply infiltrating endometriotic (DIE) lesions, in ovarian endometriomas, and in the eutopic endometrium of the same endometriosis patients and to compare their expression with that in the eutopic endometrium of women without endometriosis. To the best of our knowledge, the expression of IGF-1 isoforms, particularly in DIE cases, has not been studied thus far. We hypothesized that IGF-1 isoforms may exhibit different expression profiles in the different forms of endometriosis, affecting their phenotype, aggressiveness, and histological features.

## 2. Materials and Methods

### 2.1. Study Design and Sample Collection

The current study included 39 patients: 28 patients with surgically and histologically-proven endometriosis, of whom 15 had endometriomas without evidence of deep endometriosis (EMA-only group), 7 had deeply infiltrating endometriosis (DIE-only group), and 6 had concurrent DIE and ovarian endometriosis (EMA+DIE group), as well as 11 patients as the control group who underwent laparoscopic surgery for non-submucosal fibroids (staged as FIGO 3–6) and in whom no endometriosis was found intraoperatively. Patients with submucosal fibroids or endometrial polyps were excluded because of the potential impact of the expression of IGF-1 isoforms on the eutopic endometrium. Also, none of the patients had received any type of hormonal treatment during the last 6 months. All patients underwent surgical treatment in the Endoscopic Surgery Unit of the 1st University Department of Obstetrics and Gynecology of the National and Kapodistrian University of Athens, Greece, from 2014 to 2018.

The protocol of the current study was approved by the Ethics Committee of “Alexandra” Hospital, 1st Department of Obstetrics and Gynecology of the National and Kapodistrian University of Athens, Greece. Informed consent for both the surgical procedure and participating in the study was obtained from all participants. Demographic characteristics, medical history, clinical symptoms, preoperative work-up findings, intraoperative findings, and the final histological examination report were collected. None of the patients had received hormone treatment within 6 months before surgery. Staging of endometriosis was based on the revised American Society of Reproductive Medicine classification (r-ASRM) [13], and the original ENZIAN scoring classification system was used for patients with DIE [14].

Representative biopsies from excised ovarian and deep endometriosis lesions were obtained. The presence of endometriosis, including both epithelium and stroma, was confirmed histopathologically in sections from these specimens stained with hematoxylin and eosin. Samples of the eutopic endometrium were obtained from all patients by Pipelle endometrial biopsy at the beginning of the surgery. The excised tissue was cryopreserved at −80 °C until the time of RNA isolation.

### 2.2. Tissue Homogenization, RNA Isolation, and Quantitative PCR

Total RNA isolation was performed using the Monarch Total RNA Miniprep kit and extracted RNA was transformed into cDNA using ProtoScript II (NEB, Ipswich, MA, USA). Subsequently, real-time PCR was performed using the LightCycler 480 instrument II and LUNA Universal qPCR Master Mix (NEB) to detect the expression of different IGF-1 transcripts.

The primer set sequences used for the specific detection of IGF-1 isoforms (IGF-1Ea, IGF-1Eb, IGF1-Ec) and the housekeeping gene, glyceraldeyde-2-phosphate dehydrogenase (GAPDH), were:GAPDH-Forward(F) Sequence CAA CTC CCT CAA GAT TGT CAG CAAGAPDH-Reversed (R) Sequence GGC ATG GAC TGT GGT CAT GAIGF-1Ea F GTG-GAG-ACA-GGG-GCT-TTT-ATT-TCIGF-1Ea R CTTGTTTCCTGCACTCCCTCTACTIGF-1Eb F ATGTCCTCCTCGCACCTCTIGF-1Eb R CCTCCTTCTGTTCCCTCIGF-1Ec F CGAAGTCTCAGAGAAGGAAAGGIGF-1Ec R ACAGGTAACTCGTGCAGAGC

Each PCR reaction contained 50 ng of cDNA, 10 μL master mix, 1 μM of each primer, and nuclease-free water to a total volume of 20 μL. The real-time PCR parameters were the following: initial denaturation at 95 °C for 5 min followed by 40 cycles of 30 s at 95 °C, 30 s at 62 °C for annealing, and 30 s at 72 °C for extension. GAPDH gene expression was used as an internal control to normalize the expression levels of the genes of interest (IGF-1 isoforms). Relative quantification of IGF-1 isoform expression was analyzed using the 2(-Delta Delta C(T)) method, and relevant results are presented in diagrams as fold changes [15].

### 2.3. Statistical Analysis

Statistical analysis was performed using SPSS version 20 software.

Quantitative data are presented as mean values and range. Two-sample t-test (independent samples or paired) and their non-parametric analogs, such as Wilcoxon Signed Rank test and Mann–Whitney U test, were used for comparisons between groups.

The statistical significance cut-off was set at 5%.

## 3. Results

The mean ages (±SD) of patients were 34.3 (±6.13) and 32.8 (±5.73) years in the endometriosis and control groups, respectively. There was no significant age difference between patients with and without deep endometriosis.

The mean maximum diameter of endometriomas for the EMA-only group was 5.43 cm (ranging from 2 to 8 cm), while for the EMA+DIE group, the mean maximum diameter of endometriomas was calculated to be 4 cm (ranging from 1.5 to 6 cm).

The mean maximum diameter of deep endometriosis nodules was estimated to be 2.89 cm.

All patients had stage III–IV endometriosis according to the r-ASRM classification.

Deep endometriotic nodules were all located in the posterior compartment.

The age, size of endometrioma and location, and size of deep endometriosis nodule and location for all patients are presented in Table 1.

The expression of all isoforms of IGF-1 was significantly higher in the endometriosis group than in the control group (*p*-value IGF-1Ea: 0.001, *p*-value IGF-1Eb < 0.001, *p*-value IGF-1Ec: 0.000 < 1). IGF-1Ec, in particular, was the isoform most highly expressed in the former group (Figure 1).

When the expression of the three isoforms of IGF-1 in the eutopic endometrium of the EMA-only group was compared to that of the DIE-only group, it was found that IGF-1Ea and IGF-1Eb had no significant expression differences (*p* = 0.337, and *p* = 0.911, respectively). The expression of IGF-1Ec was 3 times higher in the DIE-only group, and this difference reached statistical significance (*p* < 0.001) (Figure 2).

A similar trend with the aforementioned findings was observed when IGF-1 isoform expression in cysts of the EMA-only group was compared with that in cysts of the DIE-only group. IGF-1Ea expression was higher but not significantly so in the latter group (*p* = 0.50), whereas IGF-1Eb expression was similar (*p* = 0.962). On the contrary, IGF-1Ec expression was almost 9 times higher in the DIE-only group, a statistically significant level (*p* < 0.001) (Figure 3).

When IGF-1 isoform expression in the eutopic tissue of the EMA-only group was compared to that in cysts of the same group, it was found that all isoforms exhibited significantly higher expression in cysts than in the eutopic endometrium (*p*-value IGF-1Ea: 0.025, *p*-value IGF-1Eb: 0.028, *p*-value IGF-1Ec: 0.064). IgF-1Eb was the isoform with the highest level of expression compared to the other two, whereas the observed expression of IGF-1Ec was not significantly higher (Figure 4).

In the DIE-only group, all IGF-1 isoforms were highly expressed in both types of endometriotic lesions compared with the eutopic endometrium (*p*-value IGF-1Ea: 0.025, *p*-value IGF-1Eb: 0.028, *p*-value IGF-1Ec: 0.064). Endometriotic cysts of this group showed higher expression of both IGF-1Ea and IGF-1Ec (the highest) in comparison to that in DIE nodules (*p*-values < 0.001). IGF-1Ea and IGF-1Ec appeared to exhibit a gradual increase in their expression moving from the eutopic endometrium to DIE nodules to endometriotic cysts. IGF-1Eb had a similar expression profile in these two lesion types (Figure 5).

## 4. Discussion

Despite extensive research and the proposal of novel molecular mechanisms in endometrial diseases [16], the pathogenesis of endometriosis has yet to be determined. Several immunological, growth, and angiogenetic factors have been implicated in the establishment, survival, and development of endometriotic lesions. IGF-1 is known to be one of the factors that prevents apoptosis and acts as a mitogen on endometrial cells, resulting in their growth in the peritoneal cavity [17,18].

To the best of our knowledge, the current study is the first to examine the expression of all IGF-1 isoforms in endometriosis cases, including those with DIE nodules. We noticed a similar pattern of expression of the IGF-1Ea and IGF-1Ec isoforms in endometriotic lesions, which differed from that of IGF-1Eb. We also observed a tendency of lower expression of IGF-1Ea and IGF-1Ec in the endometriomas of patients without DIE compared to endometriomas developing in patients with concurrent DIE and to that in DIE nodules. Our results are consistent with our primary hypothesis that differential expression of IGF-1 isoforms in the different endometriosis subtypes may indicate that DIE and simple ovarian endometriosis should be considered as lesions developing under different pathogenetic mechanisms. Additionally, endometriomas developing in patients with concurrent DIE nodules appeared to exhibit a different molecular (i.e., more aggressive) phenotype in comparison with endometriomas in non-DIE cases.

The above findings are also consistent with clinical observations made by our team and others [19,20,21]. Endometriomas found in patients with DIE nodules are frequently more difficult to dissect, are associated with increased fibrosis and vascularity of the surrounding ovarian stroma, and have surgical planes that are more difficult to recognize and follow. This behavior is probably related to different molecular alterations than in endometriomas in non-DIE cases that follow, to a certain extent, the molecular pattern observed in their co-existing DIE nodules.

The fact that endometriomas in DIE patients showed higher expression of both IGF-1Ea and IGF-1Ec in comparison with DIE nodules was also an interesting finding. Different patterns of expression between these two different types of lesions probably indicates that the local microenvironment significantly affects cellular growth. In general, DIE nodules have a slower rate of growth, and this can be largely explained by their histological composition, i.e., endometriotic microcysts harbor the epithelial component, which rests in the core of the lesion, surrounded by fibrosis and smooth muscle hyperplasia [21,22,23]. Their relative distant location from the ovary deprives these lesions of a high local ovarian steroid concentration. Endometriomas, on the other hand, develop in an estrogen-rich environment that fuels their growth, and this may be reflected by a higher expression of growth factors. Nevertheless, it became evident from our study that molecular changes that are similar to those in endometriotic lesions (ovarian and deep) also exist—although to a lesser extent—in the eutopic endometrium of patients with DIE. Such changes in the eutopic endometrium of DIE patients are more pronounced in comparison to those in the eutopic endometrium of EMA-only cases. This could indicate that the eutopic endometrium has a different potential and possesses a more aggressive molecular phenotype in patients with deep endometriosis compared with those without deep lesions. As a logical consequence, ovarian endometriosis developing in these two groups follows different molecular pathways, which are directed by different earlier molecular changes with a genetic and/or epigenetic character.

Our findings support the genetic-epigenetic theory for the pathogenesis of endometriosis that has been recently proposed [24]. According to this theory, the original cell developing in an endometriotic lesion can be an endometrial cell, a stem cell, or a bone marrow cell, with their inherited genetic and epigenetic defects. These defects, together with additional acquired defects without expression, constitute the cell’s predisposition. After implantation or metaplasia, defined as stable and transmittable changes, subtle and microscopic lesions occur. Additional genetic or epigenetic changes are required for these cells to change behavior and progress into typical, cystic, deep, or other lesions [24]. Based on this theory and our findings, it appears logical to accept that the eutopic endometrium in DIE patients has a different behavioral and developmental potential when transferred to an ectopic location, for example via retrograde menstruation, than the eutopic endometrium of non-DIE patients.

Previous research has indicated that all endometriosis subtypes are clonal, as demonstrated for typical [25], deep [26], and cystic ovarian [27,28,29] lesions. Multifocal, monoclonal lesions in one woman may derive from different progenitor cells [25]. These observations are also in accordance with our findings. In DIE cases, ovarian and deep lesions exhibit similar expression tends for all IGF-1 isoforms, and this expression is different from that in endometriotic cysts of the EMA-only group. Furthermore, the genetic/epigenetic theory makes it conceivable that the fibrosis surrounding deep endometriosis lesions and eventually the outer cell layers might be composed of normal cells with reversible “metaplastic” changes induced by the endometriosis lesion through cell–cell interactions [30,31]. This previous finding may explain the fact that nodules show less IGF-1 isoform expression than endometriotic cysts in the group with concurrent lesions. Sampling of DIE nodules after their resection may also have played some role in these differences.

In this study, IGF-1Ec was the isoform most highly expressed in endometriotic lesions in DIE patients, both ovarian and infiltrative. Previous studies have provided evidence that local production of IGF-1 requires isoform-specific (extension) E peptides to drive hypertrophy in growing skeletal muscle and that both common and unique pathways exist for IGF-1 isoforms to promote biological effects [11,31]. There has been increasing interest in the differential expression and implication of IGF-1 isoforms in the regulation of muscle fiber regeneration and hypertrophy following mechanical overloading and damage [32]. Interestingly, however, there is no study to date providing insight on the role of IGF-1 isoforms in the development of smooth muscle hypertrophy and hyperplasia, especially in the context of deep infiltrative endometriotic nodules.

We previously revealed that the glandular cells of the eutopic endometrium did not express any of the IGF-1 isoforms; however, the glandular cells of the ectopic endometrium (red lesions) did express IGF-1Ec at the mRNA and protein levels. In addition, using an in vitro endometriosis model, we found that the synthetic Ec peptide of the IGF-1Ec isoform stimulated the growth of KLE endometrial-like cells, suggesting that the IGF-1Ec isoform might generate, apart from the IGF-1 receptor-mediated bioactive mature IGF-1 peptide, another post-translational bioactive product that may have an important role in endometriosis pathophysiology [33]. In this study, IGF-1Ec was the isoform most highly expressed in endometriotic lesions in patients with DIE, both ovarian and infiltrative. The finding of higher expression of the IGF-1Ec isoform in lesions in the DIE group may suggest that smooth muscle hyperplasia and possibly increased fibrosis that surrounds the active epithelial component may be derived via a certain pathway involving this isoform. Such molecular pathways may operate through the process of epithelial–mesenchymal transition (EMT), a multi-stage process wherein epithelial cells undergo changes in cytoskeleton, apical–basal polarity, and cell-to-cell contact in order to acquire a mesenchymal cell phenotype with expression of mesenchymal markers [34,35] (Figure 6). Interestingly, endogenously produced IGF-1Ec was found to induce cellular proliferation in human prostate cancer cells in vitro and in vivo by activating the ERK1/2 pathway, while tumors and cells overexpressing the Ec peptide of the IGF-1Ec isoform presented evidence of epithelial-to-mesenchymal transition [34,35]. In endometriosis, EMT is recently gaining ground as an important step in the establishment of ectopic lesions. It is characterized by the appearance of intermediate cell states with hybrid epithelial/mesenchymal phenotypes [34,35,36]. We could assume that this process holds a rather supportive and balancing role that enables endometriotic cells to resist homeostatic mechanisms driving ectopic cells to apoptosis, and in the case of deep lesions, facilitates cell survival and growth in an environment of lesser estrogenic support.

Hypoxia and an estrogen-rich environment are two stimulating signals acting on endometriotic lesions that may contribute to the EMT process. Hypoxic conditions trigger the overexpression of hypoxia-inducible factors, which are associated with EMT cellular changes [35,37]. On the other hand, endometriotic lesions are associated with higher estradiol biosynthesis and lower estradiol inactivation [38]. It is possible that in deep nodules, hypoxia is the predominant driving force acting on growth factor-dependent pathways, whereas a local estrogen-rich environment is more likely to be involved in the development of ovarian lesions in DIE patients. Local environmental stress may also play an important role in the establishment of non-reversible changes, as suggested by the genetic-epigenetic theory.

Concerning the current evidence regarding the role of IGF-1 in endometriosis, conflicting results exist. It should be stressed that the potential differential expression of IGF-1 isoforms in the different subtypes of endometriosis has not been studied to date. Furthermore, no distinction among endometriotic lesions is commonly made between deep infiltrative and other. This, as it has been clearly shown in our study, may be of vital importance. Additionally, the type of material sampled (peritoneal fluid, blood, tissue), the method of endometriotic tissue sampling, preservation (archival vs. fresh frozen), and handling, and the analysis technique are very important factors that may have played a decisive role in the results and interpretation of the findings for each study.

Several studies have shown increased activity of IgF-1 in the peritoneal fluid of women with endometriosis. Yan Zhou et al. concluded that IGF-1 stimulates estrogen (ERβ) and aromatase overexpression from endometriotic stromal cells through complex transcriptional pathways, resulting in the progression and maintenance of endometriosis [39]. In addition, Forster R. et al. showed that IGF-1 concentrations were elevated in the peritoneal fluid of patients with endometriosis compared to those without endometriosis [40]. Heidari S. et al. showed that women with endometriosis had increased IGF-1 expression in both the serum and peritoneal fluid when compared to controls [8]. Kim J.G. et al. reported that IGF-1 levels in the peritoneal fluid were significantly higher in patients with endometriosis than in control patients [17].

On the contrary, evaluation of IGF-1 activity in tissues or serum have provided conflicting results. Sbracia et al. showed more intense staining during the proliferative phase of both stromal and epithelial cells by applying immunohistochemistry for IGF-1 in controls. Specifically, in the eutopic endometria of women with endometriosis, a reduction in staining was observed, whereas in the epithelial cells of fibrotic peritoneal adhesions, intense immunostaining for IGF-1 was observed [41]. Milingos et al. observed a significant decrease in IGF1 expression in endometriotic cysts in comparison to the eutopic endometrium of women with ovarian endometriosis [12]. Matalliotakis et al. concluded that IGF-1 soluble levels were not affected in either healthy or endometriotic subjects [42]. Steff A.M. et al. found no significant differences in serum levels of IGF-1 between cases and controls in the luteal phase of the cycle [43]. Gurgan et al. found that the mean serum and peritoneal IGF-1 levels of controls and early-stage endometriosis cases were significantly lower than those of late-stage cases [44].

Furthermore, insulin-like growth factors (IGFs) have been shown to play critical roles in modulating gonadotrophin-mediated folliculogenesis and steroidogenesis. Cuhna Filho et al. analyzed the levels of insulin-like growth factor-1 (IGF-1) and IGF-binding protein-1 and -3 (IGFBP-1 and IGFBP-3) in the follicular fluid environment of infertile patients with endometriosis and concluded that the expression of both proteins was not significantly different among the groups [45].

By reviewing the existing literature regarding the possible role of IGF-1 on the pathogenesis and pathophysiology of endometriosis, it is evident that this important growth factor remains largely understudied. It appears that its isoforms may play different roles in the different forms of the disease, potentially influenced by the local microenvironment and pre-existing and superimposed genetic and epigenetic molecular alterations. Further studies are needed to clarify its significance in the development, survival, and remodeling of endometriotic lesions.

The current study aims to shed light on the expression as well as potential role of IGF-1 isoforms in endometriosis, a field that has been poorly investigated thus far. Although the number of patients included in the study could be considered as a limitation, our study comprises one of the most valuable databases on this specific topic. Amongst the strengths, we could highlight that we have assessed the level of expression of all three IGF-1 isoforms in actual, surgically excised, endometriotic tissue of all endometriosis subtypes. To our knowledge, this is the first time that the expression of IGF-1 has been studied in deep infiltrative endometriotic lesions and compared to that in other endometriosis subgroups.

## 5. Conclusions

The current study is the first to our knowledge to identify the expression of all IGF-1 isoforms in DIE samples. We noticed a similar pattern of expression between IGF-1Ea and IGF-1Ec, which differed from that of the IGF-1Eb isoform, possibly implying differential biological actions of the different isoforms in DIE subtypes. We observed a tendency of lower expression of IGF-1Ea and IGF-1Ec in endometriomas without DIE compared to endometriomas with concurrent DIE or in DIE nodules. Our results are consistent with our primary hypothesis that differential expression of IGF-1 isoforms may indicate that DIE with its associated ovarian lesions and simple ovarian endometriosis should be considered as two forms of the disease developing under different molecular pathways. The presence of endometriomas in patients with concurrent deep nodules is suggestive of an advanced and aggressive disease wherein both lesions may follow similar growth pathways involving the IGF-1Ec isoform, whereas DIE-only cases are also characterized by the active and aggressive nature of these lesions, exhibiting similar, but to a lesser degree, increased IGF-1Ec expression. Increased expression of IGF-1 in the EMA+DIE group compared to that in the DIE-only group may be consistent with current knowledge that ovarian endometrioma is indeed a marker for greater severity of DIE.

## Figures and Tables

**Figure 1 biomolecules-14-00007-f001:**
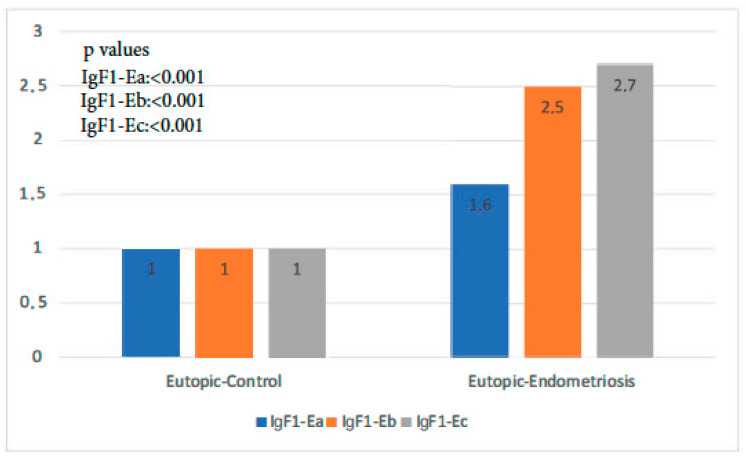
IGF-1 isoform expression in eutopic endometrium of control group versus endometriosis group (all cases).

**Figure 2 biomolecules-14-00007-f002:**
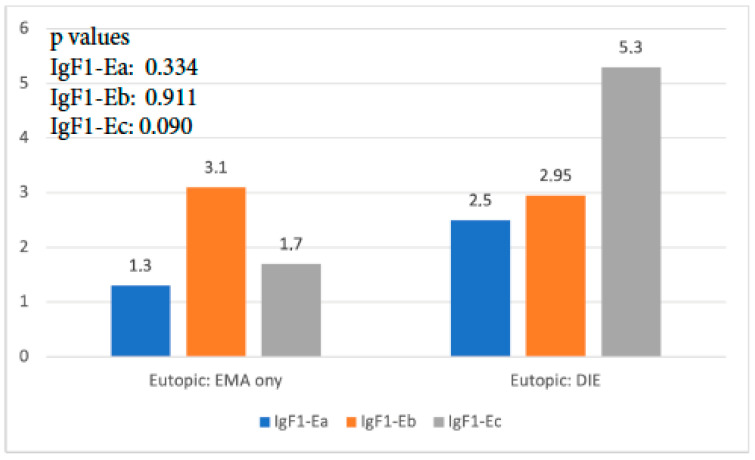
IGF-1 isoform expression in eutopic endometrium of EMA-only group versus DIE-only group.

**Figure 3 biomolecules-14-00007-f003:**
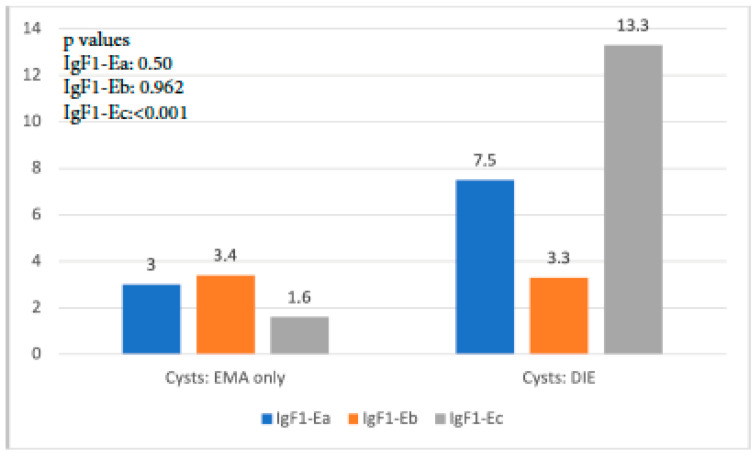
IGF-1 isoform expression in endometriotic cysts of the EMA-only group versus that in cysts of the DIE-only group.

**Figure 4 biomolecules-14-00007-f004:**
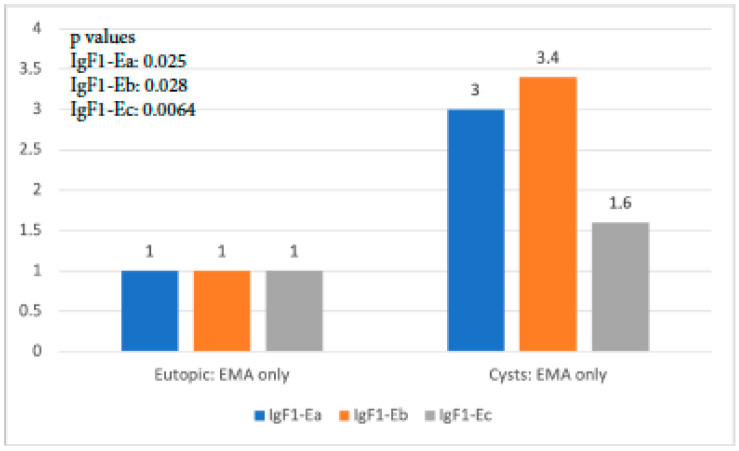
IGF-1 isoform expression in eutopic endometrium versus endometriotic cysts in the EMA-only group.

**Figure 5 biomolecules-14-00007-f005:**
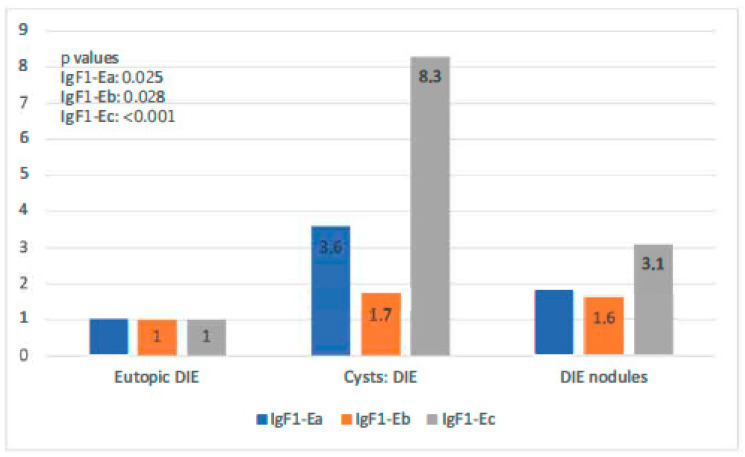
IGF-1 isoform expression in eutopic endometrium versus endometriotic cysts versus DIE.

**Figure 6 biomolecules-14-00007-f006:**
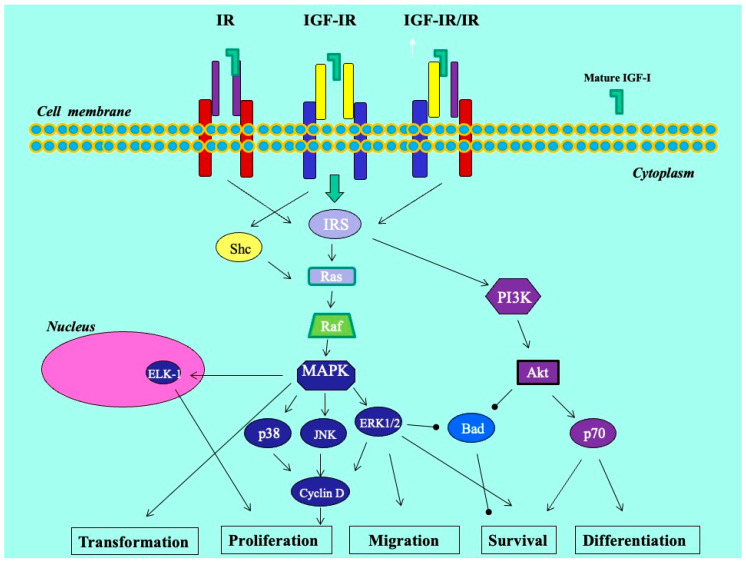
Molecular mechanisms in endometriosis.

**Table 1 biomolecules-14-00007-t001:** Endometriosis patient characteristics.

Pt	Age	Size (cm) and Location of Endometrioma		Size (cm) and Location of DIE Nodule	
		Right Ovary	Left Ovary		
Endometrioma only patients (EMA-only group)					
1	31	5	2	(-)	
2	28	3 + 4	(-)	(-)	
3	31	5	(-)	(-)	
4	26	8 + 7	(-)	(-)	
5	31	(-)	3	(-)	
6	35	7	5	(-)	
7	45	7	(-)	(-)	
8	31	7	(-)	(-)	
9	38	7	(-)	(-)	
10	29	(-)	4	(-)	
11	34	(-)	8	(-)	
12	35	7	(-)	(-)	
13	36	4	2	(-)	
14	32	(-)	6	(-)	
15	44	5	5	(-)	
Endometrioma and DIE patients (EMA+DIE group)					
16	24	1.4	4	3	PVW
17	38	4	(-)	1.8	Rectum
18	29	3	5	2.5	RVS
19	41	(-)	5	1.8	RVS
20	38	(-)	6	5.5	RVS
21	36	(-)	3.5	2.6 + 3.2	R USL/L USL
DIE only patients (DIE-only group)					
22	29	(-)	(-)	2.5	R USL infiltrating PVW
23	36	(-)	(-)	3.5	L USL
24	36	(-)	(-)	4	POD
25	29	(-)	(-)	3 + 1.3	L USL/Rectum
26	42	(-)	(-)	3	R USL
27	30	(-)	(-)	3 + 2.2	POD/R USL
28	46	(-)	(-)	3	L USL

USL: uterosacral ligament, POD: pouch of Douglass, RVS: rectovaginal septum, PVW: posterior vaginal wall.

## Data Availability

Data are available upon request.

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
