# Peer review of "Differential Expression of Insulin Growth Factor 1 (IGF-1) Isoforms in Different Types of Endometriosis: Preliminary Results of a Single-Center Study"

_biomolecules, 2023, doi:10.3390/biom14010007_

Round 1

Reviewer 1 Report

Comments and Suggestions for Authors

I read with great interest the manuscript, which falls within the aim of this Journal and offers a high-quality overview of the topic.

In my honest opinion, the topic is interesting enough to attract the readers’ attention. The abstract perfectly summarizes the contents of the manuscript. The introduction is satisfactory. Methodology is accurate and conclusions are supported by the data analysis. The tables and figures are clear and interesting.

Although the manuscript can be considered already of high quality, I would suggest taking into account the following minor recommendations:

- I suggest another round of language revision, in order to correct a few typos and improve readability.

- Inclusion/exclusion criteria should be better clarified by extending their description.

- Discussions can be expanded and improved by citing relevant articles (I suggest authors to read and insert in references the following article PMID: 36979434).

- The authors have not adequately highlighted the strengths and limitations of their study. I suggest better specifying these points.

Comments on the Quality of English Language

Minor editing of the English language is required to make the work clearer and more readable.

Author Response

Dear Editor/Reviewers,

We would like to express our gratitude for your invaluable feedback and insightful comments on our manuscript entitled ‘’Differential Expression of Insulin Growth Factor 1 (IGF-1) Isoforms in Different types of Endometriosis. Preliminary results of a Single-center Study. ‘’, which is considered for publication in your special Issue of Novel Insights into Molecular Mechanisms of Endometrial Diseases.

Following up on the recommendations made by all three reviewers, we have made several key changes. Please find attached the docx file‘’Biomolecules-2742591_Revised’’, where all the changes have been highlighted by added comments.

Specifically:

  • Spacing and typo mistakes have been checked throughout the whole manuscript and appropriate changes have been made.
  • As per reviewer’s 2 advice, the word ‘angiogenic’ has been replaced by ‘angiogenetic’, and ‘’3’’ has been removed from line 131.
  • Inclusion and exclusion criteria clarification (lines 69-78|): Endometriosis group consists of patients with surgically and histologically proven endometriosis, whereas control group includes patients with specific type of fibroids (only FIGO type 3-6 fibroids). We have excluded any patients with submucosal fibroids or endometrial polyps, because of their potential impact on the expression of IgF1 isoforms of the eutopic endometrium, as well as patients that have taken any form of hormonal therapy during the last 6 months.
  • Limitations and strengths of our study have been added in lines 395-402.
  • Y- axis of all diagrams can be labelled as ‘’Fold change’’. Relevant reference [15] (PMID: 11846609) has been added in line 123-125.
  • Reference [16] (PMID:3697943) has been added in lines 190-191.
  • Please note that the references numbers and relevant listing have been updated.
  • A comment about the difference regarding the surgical treatment of endometriomas in EMAs-only compared to EMA+DIE group has been documented in lines 209-211, while the variation in terms of size and location of endometriotic lesions between the groups are presented in Table 1.
  • Correlating our findings with patient’s symptoms and additional patient characteristics (i.e. reproductive outcome, comorbidities) is a great idea. We recognize the significance of this correlation for further understanding the implications of our study. We are currently actively pursuing this avenue as a separate, dedicated project and we are planning to conduct a new statistical analysis to reach reasonable conclusions.

Once again, we sincerely appreciate the time and the effort you dedicated to reviewing our manuscript.

Your feedback has been instrumental in refining our work. We firmly believe that the revisions made have significantly strengthened the manuscript. We hope that these changes meet your expectations and look forward to your further guidance.

Thank you for your continued support and consideration.

Kind regards,

Nikolaos Blontzos

Reviewer 2 Report

Comments and Suggestions for Authors

Very minor language changes are recommended such as the use of the words "angiogenic" and angiogenetic".  While both are correct for consistency please use one version.  There are a few locations where spacing needs correction and on Line 120 there is a "3." which should be removed as it is out of place.

Author Response

(The authors gave the same response as above.)

Reviewer 3 Report

Comments and Suggestions for Authors

The authors of this study aimed to analyze the differential expression of insulin-like growth factor 1 (IGF-1) isoforms in different types of endometriosis lesions. The authors suggest that the differential expression of IGF-1 isoforms may indicate that deeply infiltrating endometriosis (DIE) with its associated ovarian lesions and simple ovarian endometriosis should be considered as two forms of the disease developing under different molecular pathways. While the manuscript is well written, there are a few major issues that require the authors’ attention:

The Y-axis of the figures is not labeled.

Discuss the clinical significance of the differences in the mean maximum diameter of endometriosis between the EMA-only and EMA+DIE groups. Does the size correlate with symptom severity, disease progression, or treatment considerations?

While it's mentioned that all patients had stage III-IV endometriosis according to the r-ASRM classification, discuss the clinical significance of this staging in the context of the study objectives.

Provide insights into how the characteristics (size and location) of endometriotic lesions vary between the EMA-only and DIE groups. Discuss any implications for the understanding of disease heterogeneity.

Elaborate on the clinical significance of all deep endometriotic nodules being located in the posterior compartment. Does this have any implications for symptomatology or surgical management?

Consider providing additional patient characteristics relevant to the study objectives, such as reproductive history, symptoms, or any comorbidities that might influence the expression of IGF-1 isoforms.

Author Response

(The authors gave the same response as above.)

Round 2

Reviewer 3 Report

Comments and Suggestions for Authors

The authors have done a good job with the revision. I support this publication.

Author Response

We are excited that you find our manuscript appropriate for publication